Zero-shot cross-lingual stance detection via adversarial language adaptation

A. Bharathi 1
http://orcid.org/0000-0003-4583-3623 Zubiaga Arkaitz 2 a.zubiaga@qmul.ac.uk
1 Department of Applied Mathematics and Computational Sciences, PSG College of Technology , Coimbatore , India
2 Queen Mary University of London , London , United Kingdom
Alatas Bilal
Electronic publication date: 2025 Jul 3
Publication date: 2025
Volume: 11
Electronic Location ID: e2955
Received 2024 Oct 4; Accepted 2025 May 22
Copyright: © 2025 A. and Zubiaga
Copyright year: 2025
Copyright holder: A. and Zubiaga
License: This is an open access article distributed under the terms of the Creative Commons Attribution License, which permits unrestricted use, distribution, reproduction and adaptation in any medium and for any purpose provided that it is properly attributed. For attribution, the original author(s), title, publication source (PeerJ Computer Science) and either DOI or URL of the article must be cited.
License URL: https://creativecommons.org/licenses/by/4.0/

Keywords: Cross-lingual stance detection, Zero-shot stance detection, Multilingualism, Cross-lingual NLP, Stance detection, NLP, Transfer learning, Deep learning

Funding: The authors received no funding for this work.

==============================
Stance detection has been widely studied as the task of determining if a social media post is positive, negative or neutral towards a specific issue, such as support towards vaccines. Research in stance detection has often been limited to a single language and, where more than one language has been studied, research has focused on few-shot settings, overlooking the challenges of developing a zero-shot cross-lingual stance detection model. This article makes the first such effort by introducing a novel approach to zero-shot cross-lingual stance detection, multilingual translation-augmented bidirectional encoder representations from Transformers (BERT) (MTAB), aiming to enhance the performance of a cross-lingual classifier in the absence of explicit training data for target languages. Our technique employs translation augmentation to improve zero-shot performance and pairs it with adversarial learning to further boost model efficacy. Through experiments on datasets labeled for stance towards vaccines in four languages—English, German, French, Italian, we demonstrate the effectiveness of our proposed approach, showcasing improved results in comparison to a strong baseline model as well as ablated versions of our model. Our experiments demonstrate the effectiveness of model components, not least the translation-augmented data as well as the adversarial learning component, to the improved performance of the model. We have made our source code accessible on GitHub: https://github.com/amcs18pd05/MTAB-cross-lingual-vaccine-stance-detection-2.

Introduction

Stance detection is a widely studied natural language processing (NLP) task that consists in determining the viewpoint expressed in a text towards a particular target (Küçük & Can, 2020; AlDayel & Magdy, 2021). By determining the viewpoints of a large collection of posts collected through online sources such as social media, one can then get a sense of the public opinion towards the target in question (Cao et al., 2022; Almadan, Maher & Windett, 2023). Given a text as input, a stance detection model then determines if it conveys a supporting (positive), opposing (negative) or neutral viewpoint towards a target. Recent research has therefore looked into improving stance detection models that make competitive predictions across all three classes, in domains such as politics (Liu et al., 2022) and healthcare (Conforti et al., 2020).

In line with recent progress in NLP research (Min et al., 2023; Zubiaga, 2024), state-of-the-art stance detection models embrace methods based on large language models such as Transformers (Karande et al., 2021; Gunhal et al., 2022; Zhao, Li & Caragea, 2023). However, where stance detection is of global interest, research has generally studied the applicability of models to a single language, predominantly English, and generalizability to other languages has been understudied. Stance detection models, primarily trained on a single language, face challenges when applied to multilingual scenarios (Lai et al., 2020). The paucity of labeled data for each language further complicates this task, demanding innovative strategies to navigate linguistic diversity effectively, particularly in the case of low-resource languages where labeled data is scarce or unavailable. While limited prior research has explored cross-lingual methods for stance detection (Hardalov et al., 2022; Mohtarami, Glass & Nakov, 2019), there has been little effort toward developing zero-shot cross-lingual approaches that can be applied when no labeled data is available in the target language. Zero-shot cross-lingual stance detection refers to the task of identifying the stance (e.g., support, oppose, neutral) of a text in a target language without having seen any labeled training data in that language. Zero-shot learning relies solely on knowledge transferred from a high-resource language (e.g., English) to make predictions in low-resource languages. This is particularly important for lesser-resourced languages, where labeling relevant datasets is often impractical or cost-prohibitive (Cao et al., 2019; Ding et al., 2020; Hedderich et al., 2021).

In this work, we present the first method for zero-shot cross-lingual stance detection, moving away from prior approaches in few-shot settings that rely on a small amount of labeled data from the target language during training. To achieve this, we propose multilingual translation-augmented bidirectional encoder representations from Transformers (BERT) (MTAB), a novel architecture that integrates three core components: (i) translation augmentation, (ii) a multilingual encoder, and (iii) adversarial language adaptation.

Translation augmentation refers to the process of augmenting the training data by translating source-language instances into multiple target languages using a machine translation system. Formally, given a labeled instance (xs,y) in source language Ls, we generate a set of translated instances {(Ts→ti(xs),y)}i=1k, where Ts→ti is a translation function from Ls to a target language Lti. This encourages the model to learn language-invariant representations by exposing it to diverse linguistic realizations of the same semantic content.

Adversarial language adaptation is a domain adaptation technique that aligns feature representations across languages using adversarial training. Let fθ(x) denote the shared encoder’s output representation for input x, and let Dϕ be a language discriminator trained to predict the language of fθ(x). The encoder is trained to minimize the stance classification loss Ltask while simultaneously maximizing the discriminator loss Ladv, i.e., minimizing the objective Ltask−λLadv. This adversarial objective ensures that the encoder learns language-agnostic features, improving generalization to unseen languages.

The multilingual encoder in MTAB is based on a pretrained multilingual transformer model (e.g., mBERT or XLM-R), capable of processing multiple languages by learning shared contextual embeddings. By combining translation augmentation with adversarial alignment, MTAB enables effective zero-shot transfer to new languages without any labeled data from the target language during training.

To test the effectiveness of MTAB, we perform experiments on datasets in four different languages (i.e., English, French, German and Italian) for the same domain, namely stance detection around vaccine hesitancy. Our experiments show that this combination consistently outperforms cross-lingual transfer in models that only use adversarial learning as a technique to align embeddings to the target languages. Our research establishes a benchmark opening up a novel research direction into zero-shot cross-lingual stance detection.

Our work makes the following novel contributions: We introduce MTAB, a novel approach to zero-shot cross-lingual stance detection that uses a translation-augmented training dataset and an adversarial learning model to adapt to unseen languages with no labelled examples. We believe that our approach is not only applicable to this particular task, but can also be extended to other cross-lingual text classification problems.

To the best of our knowledge, ours is the first work addressing zero-shot cross-lingual stance detection. Additionally, it is the first study to apply an adversarial learning approach to cross-lingual stance detection.

Experimenting across four languages (English, French, German, Italian) in the domain of vaccine hesitancy, we demonstrate the effectiveness of MTAB for cross-lingual stance detection. Through further ablation experiments, we prove the contribution of each of the components of MTAB towards improved performance.

Cross-lingualism remains a less-explored topic in stance detection, and our research aims to bridge that gap. As one of the first studies in this area, we delve into the potential of standard NLP and transfer learning techniques to tackle cross-lingual stance detection, contributing to the evolving understanding of this domain.

The remainder of the article is organized as follows. Next, in “Related Work”, we discuss prior research relevant to ours and we highlight our key novel contributions on top of those. We then describe the datasets we use in our study in “Datasets”. We introduce and describe our newly proposed model, MTAB, in “MTAB: Model Architecture”, followed by details of our experimental settings in “Experiments”. We show and analyze results of our experiments in “Analysis of Results”, subsequently concluding the article in “Conclusion”. The text of this article is taken from the preprint: Zero-Shot Cross-Lingual Stance Detection via Adversarial Language Adaptation.

Related work

The ability to accurately classify stances on unseen or unfamiliar topics or languages is of paramount importance in real-world applications, where issues and discussions are constantly evolving (Alkhalifa & Zubiaga, 2022; Alkhalifa, Kochkina & Zubiaga, 2023). Traditional stance detection models often struggle to generalize effectively across different topics, domains, and languages, limiting their practical utility (Reuver et al., 2021; Schiller, Daxenberger & Gurevych, 2021; Ng & Carley, 2022). To address this limitation, recent research has focused on developing models capable of generalizable stance detection. These efforts have involved exploring transfer learning, domain adaptation and other cross-lingual approaches to enhance the model’s ability to handle unseen or unfamiliar topics and languages.

Cross-topic and cross-target stance detection

Cross-topic stance detection aims to detect stance across topics using knowledge transfer. This approach leverages large stance datasets from source topics to improve performance on smaller target topic datasets. However, most existing work in cross-topic stance detection assumes some labelled data in the target topic is available for training. Zero-shot cross-topic stance detection aims to detect stance in an entirely novel target topic with no labelled training data, instead leveraging source topic data and topic-independent features. Though still relatively understudied, several approaches have been proposed for zero-shot cross-topic stance detection, including training on multi-target datasets, topic-invariant representation learning and unsupervised domain adaptation methods which can also be applied to scenarios involving cross-lingual and cross-domain problems.

Zarrella & Marsh (2016) proposed a system for the SemEval-2016 competition on stance detection in tweets that uses transfer learning from unlabeled datasets and an auxiliary hashtag prediction task to learn sentence embeddings. Their approach, which achieved the top score in the task, uses a recurrent neural network (RNN) initialized with features learned via distant supervision on two large unlabeled datasets, and then fine-tuned on a small dataset of labeled tweets. For the same task, Augenstein, Vlachos & Bontcheva (2016) train a Any-Target stance classifier using a bag-of-words autoencoder that learns feature representations independent of any target. Xu et al. (2018) and Wei & Mao (2019) proposed two different approaches for cross-target stance classification, both of which were able to achieve state-of-the-art results on the SemEval-2016 Task 6 test set. Xu et al. (2018) used a self-attention network to extract target-independent information for model generalization, while Wei & Mao (2019) introduced a topic modeling approach that can leverage shared latent topics between two targets as transferable knowledge to generalize across topics. Building upon these advancements, several other notable contributions have been made in achieving state-of-the-art results in cross-topic stance detection, mainly within the context of the SemEval-2016 Task 6. Wang et al. (2020) employed adversarial domain generalization to learn target-invariant text representations. In addition, Hardalov et al. (2021) combined domain adversarial learning with label-adaptive learning to learn input representation specific to the stance target domain, and handle unseen domains using embedding similarities. A more recent work by Ji et al. (2022) establishes a new state-of-the-art for the task by incorporating a model based on meta-learning, which is also trained on multiple targets considering the similarities between targets. More recently, Khiabani & Zubiaga (2023) introduced a model that leverages features derived from the social network, in addition to the textual content of posts, to improve a model for cross-target stance detection, which they showed could be further used to enhance a language model for stance detection (Jamadi Khiabani & Zubiaga, 2024).

Recent approaches in stance detection for unseen targets, or more formally zero-shot stance detection have shown promising results. One of the first works in this area by Allaway & McKeown (2020), introduced a new dataset VAST for zero shot stance detection (ZSSD) and developed a model that learns generalized topic representations to pre-dict stance labels based on similarities between given topics and the topics present in training set. Allaway, Srikanth & McKeown (2021) also proposed a new TOpic ADversarial Network (TOAD) that conditions document representations on topics and further learns domain-invariant features using adversarial training for zero-shot stance detection. Liang et al. (2022) proposed a novel approach for ZSSD that frames the distinction of target-invariant and target-specific stance features as a pre-text task to better learn transferable stance features. In the case of Twitter, an unsupervised zero-shot stance detection framework Tweet2Stance, proposed by Gambini et al. (2022) uses content analysis of users’ Twitter timelines with an NLI based zero-shot classifier to infer their stance towards a political-social statement.

Cross-lingual stance detection

Current approaches to stance detection are well-studied in English but have received less attention in other languages and cross-lingual settings. A novel approach to improve cross-lingual stance detection introduced by Mohtarami, Glass & Nakov (2019) uses memory networks along with a contrastive language adaptation component to effectively align source and target language data with the same labels and separate the ones with different labels. To facilitate research in multilingual stance detection, Zotova et al. (2020) introduced a new Twitter dataset for the Catalan and Spanish languages annotated with Stance towards the independence of Catalonia. X-stance is another multilingual stance detection dataset introduced by Vamvas & Sennrich (2020) that includes text in German, French and Italian, covering a wide range of topics in Swiss politics. A more recent work by Hardalov et al. (2022) presents a comprehensive study of cross-lingual stance detection, covering 15 datasets in 12 languages that demonstrates the effectiveness of pattern-based training approaches and pre-training using labeled instances from a sentiment model.

Existing research on cross-lingual stance detection has however been limited to few-shot settings, where a small portion of labeled data is available for the target language. In our work, we are interested in going further by enabling zero-shot cross-lingual stance detection where there is no labeled data at all for the target language. With this objective in mind, we propose our novel model MTAB, which leverages adversarial learning through translation-augmented training data for zero-shot cross-lingual stance detection.

Vaccine stance detection

The task of vaccine stance detection is still relatively new, but there has been significant progress in recent years, given the rise of vaccine misinformation on social media. Twitter is a rich source of data for vaccine discourse, and many researchers have used Twitter data to develop mono-lingual and cross-lingual vaccine stance detection models. The method introduced by Bechini et al. (2021) particularly demonstrates the importance of incorporating user related information in the analysis of public opinion on Twitter and combine that with text classification techniques to asses opinions on vaccination topic in Italy. In the month following the announcement of the first COVID-19 vaccine, Cotfas et al. (2021) provided insights into the dynamics of opinions on COVID-19 vaccination based on tweets and compared classical machine learning and deep learning algorithms for stance detection. In a related study conducted on the same period, the authors examined vaccine hesitancy discourse on Twitter using several text analysis techniques such as linear discriminant analysis (LDA), Hashtag and N-gram analysis (Cotfas, Delcea & Gherai, 2021). Addressing this challenge in low-resource languages, Küçük & Arıcı (2022) introduced the first dataset for Turkish tweets that is annotated with stance and sentiment labels towards COVID-19 vaccination. In addition, the first manually annotated Arabic tweet dataset related to the COVID-19 vaccination campaign (ArCovidVac) proposed by Mubarak et al. (2022) consists of stance annotated tweets from different Arab countries and delivers key insights into the discussions and opinions related to the COVID-19 vaccination campaign on social media in the Arab region.

The domain of vaccine hesitancy presents an ideal scenario for our research, as it is an issue of global interest, which is discussed and opinionated across languages in the world, which has in turn led to the creation of labeled datasets in different languages. In our case, we use datasets in four languages, namely English, French, German and Italian, to enable our zero-shot cross-lingual stance detection study.

Datasets

In this section, we provide a detailed description of the datasets used for training and testing our model, along with the associated pre-processing techniques. Specifically, we use English data only for training, and then use three different test sets in three different languages for testing, i.e., French, German and Italian. The statistics of our datasets for both training and testing are provided in Table 1.

Table 1 Dataset statistics.

We use the aggregated training data for training our models, which contains tweets only in English. We test our models on tweets in unseen languages individually, i.e., French, German and Italian.

Training set	
Data source	Label distribution	Total # of Tweets	
Positive	Negative	Neutral	
(A) Statistics of the training data	
Vaccine attitude dataset (VAD) by Zhu et al. (2021)	1,367	517	162	2,046	
Almadan et al. (2022)	30	12	13	55	
Cotfas et al. (2021)	879	612	901	2,392	
Aggregated training data	2,276	1,141	1,076	4,493	
Testing set—VaccinEU	
Target language	Label distribution	Total # of Tweets	
Positive	Negative	Neutral	
(B) Statistics of the testing data	
French	419	135	279	833	
German	547	108	169	824	
Italian	314	151	458	923	

Training datasets

We experiment in the scenario where a model is only trained on a resource-rich language, English, to then test it on other languages. As such, our training data only contains data in the English language, while test data will be written in other languages. We therefore set up the scenario where the target languages found in the test sets are not seen during training.

To achieve this objective and create our English-only training data, we combined data from three different sources of English tweets on COVID-19 vaccines belonging to different timelines during the period of 2020 and 2021 to create a more diverse and representative sample. We aggregate the following three datasets: (1) Vaccine Attitude Dataset (VAD) by Zhu et al. (2022): The VAD dataset (Zhu et al., 2022) consists of an extensive collection of 1.9 million English tweets obtained between February 7th and April 3rd, 2021. The authors annotated a random subset of 2,800 tweets from this corpus with stance labels and aspect characterizations. We use this smaller, labeled subset as part of our training data.

(2) COVID-19 Vaccination Tweets dataset by Almadan et al. (2022): The work of Almadan et al. (2022) presented a systematic methodology for annotating tweets with respect to their stance toward COVID-19 vaccination. This approach included the development of a comprehensive codebook for stance annotation and the use of keyword and hashtag based sampling techniques for Twitter data. Leveraging this coding framework, the authors released a stance annotated dataset for research purposes (Almadan et al., 2022), which we also incorporate into our training dataset.

(3) COVID-19 Vaccine Stance Dataset by Cotfas et al. (2021): The authors Cotfas et al. (2021) collected 2,349,659 tweets related to COVID-19 vaccination during the period of November 9th to December 8th, 2020 using a set of predefined keywords via the Twitter API. From this collection, they curated a refined and balanced stance annotated dataset comprising 3,249 tweets, which was made publicly accessible (Cotfas et al., 2021). We incorporate this smaller, labeled subset of their dataset into our training data.

After aggregating the three labeled data sources, our final training data contains 4,493 tweets, of which 2,276 are labeled positive, 1,141 negative, and 1,076 neutral. We randomly sample 90% of the aggregated data for model training and 10% for validation on English data. See Table 1A for more details on the training data, including the final distribution of labels.

Test data

For testing our model on non-English data, we use data pertaining to three languages—French, German and Italian—from the VaccinEU (Giovanni et al., 2022) dataset as introduced by Giovanni et al. (2022). To study the impact of online conversations on COVID-19 vaccines, the researchers used a list of vaccine related keywords and collected a large dataset of over 70 million tweets in three different languages, namely French, German, and Italian, from November 1st, 2020 to November 15th, 2021. From this large-scale dataset, the authors labeled small samples for each language, providing four labels—Positive, Negative, Neutral, and Out of Context. However, for our experiments, we only focus on the Positive, Negative, and Neutral classes, as detailed in Table 1B. We can observe that labels are unevenly distributed across different languages, which is itself an additional challenge for our research, but one can realistically expect that these variations across languages will naturally happen, not least because of cultural differences affecting level of support towards vaccines. Given the importance of label distributions in our cross-lingual research, we delve into performance analysis across labels in our analysis of results, in addition to looking at general performance scores aggregated for all labels.

Data pre-processing

Due to Twitter’s terms of service and privacy concerns, these data were made available only using Tweet IDs, and hence, we first used an additional application called Hydrator (https://github.com/DocNow/hydrator) to retrieve the actual tweet content associated with the IDs. Due to the age and controversial nature of the tweets, some of them were lost during retrieval. This however happens generally with Twitter datasets, as research has shown that a percentage of tweets inevitably tends to disappear (Zubiaga, 2018), and hence we conduct our research with the tweets available at the time of our collection, which are the statistics we report in Table 1.

In the first step of data pre-processing, we standardize the stance labels of the three datasets that were combined for model training. This ensures the diverse labeling conventions present in those datasets are harmonized enabling consistent and integrated experimentation.

Once the data were ready, and to prepare the tweets for further analysis and modeling, we carried out a series of textual pre-processing steps, as follows: Removal of URLs, emojis, mentions, smileys, and numbers: In the interest of focusing on the textual content of the tweets, we used the “tweet-preprocessor” (https://pypi.org/project/tweet-preprocessor/) library to eliminate URLs, emojis, user mentions, smileys, and numbers from the tweet text.

Handling retweets: Retweets often contain a standard “RT @” prefix that can affect the consistency of the text by adding noise to it which provides no information about the stance of a tweet. We removed this prefix to standardize the format of retweets while preserving the essence of their content.

Handling hashtags: To maintain the relevance of hashtags as keywords, we opted to remove only the ‘#’ symbol, retaining the keyword itself. This approach ensured that the hashtags contributed to the context of the tweet without introducing any extraneous characters, while we retain the content of the hashtag whenever this forms part of the sentence, which we did not want to break.

The resulting texts, after the three preprocessing steps above, are then fed to our models for training or testing.

Mtab: model architecture

Here we introduce the architecture of our zero-shot cross-lingual stance detection model, multilingual translation-augmented BERT (MTAB), which we design to categorize public opinions on vaccines across diverse languages, specifically targeting French, German, and Italian. Our architecture employs two levels of data augmentation combined with training a neural stance classifier on multilingual contextual embeddings that are adapted to the target languages via adversarial training. The full architecture is presented in Fig. 1, and we describe each of the three components next: (i) translation augmentation, (ii) multilingual encoder and stance classifier, and (iii) adversarial language adaptation.

Figure 1 Architecture of our proposed MTAB model.

Translation augmentation

Translation augmentation allows a model trained in one language (source language) to gain exposure to diverse linguistic contexts and structures present in other languages. We expanded our English training dataset by incorporating translations of training examples into our target languages. As mentioned in “Training Datasets”, 90% of the total source language data is used for training and the remaining 10% is used as the validation set. For obtaining translations, we used the Opus-MT (https://github.com/Helsinki-NLP/Opus-MT) model from Helsinki-NLP, accessible via the Easy-NMT (https://github.com/UKPLab/EasyNMT) package, which we chose as a competitive and open source solution. By providing translations of the training data into multiple languages, the model can learn to recognize common patterns, sentiment expressions, and stance cues that transcend linguistic boundaries. Different languages may employ distinct grammatical structures, vocabulary, and cultural nuances. By training on a multilingual dataset that includes augmented training data incorporating translations, the model learns to identify semantically equivalent expressions and sentiment cues in different languages.

This approach aligns with prior work that has successfully leveraged translation to enhance zero-shot cross-lingual transfer learning. Translation-based augmentation has been widely used in zero-shot settings to improve model adaptability without requiring human-annotated labels in the target language. Ding et al. (2022) employ translation to align embeddings of English-labeled data with target languages, facilitating more effective cross-lingual transfer using improved multilingual representations. Fazili, Agrawal & Jyothi (2024) use LLMs to generate synthetic English data for multiple NLP tasks and translate them into target languages to create relevant data sets for low-resource languages, thereby boosting zero-shot cross lingual performance. Similarly, Riabi et al. (2021) generate synthetic question-answer pairs in English and translate them into other languages, thereby augmenting the training data available for cross-lingual question answering. These studies demonstrate that machine translation can be a practical and effective strategy for mitigating the scarcity of labeled data in low-resource languages, while still adhering to the principles of zero-shot learning.

Thus, our approach remains within the zero-shot paradigm, as it does not introduce any manually labeled data in the target language. Instead, it leverages labeled data from a high-resource language to enhance model generalization across multiple languages, ensuring a fair and effective adaptation to new linguistic domains.

Multilingual encoder and stance classifier

Multilingual BERT (mBERT), pre-trained on 104 languages, has been shown to perform well for zero-shot cross-lingual transfer, especially when fine-tuned on downstream tasks. Therefore, we employed mBERT to finetune a multilingual encoder that can generate contextual embeddings tailored to our task-specific tweets. Simultaneously, we jointly trained a stance classifier, optimized to categorize stances based on these embeddings.

Adversarial language adaptation

Our approach for language adaptation was inspired by the adversarial domain adaptation (ADA) with distillation technique proposed by Ryu & Lee (2020) for unsupervised domain adaptation. Their method, ADA, aims to enhance the performance of BERT in the face of domain shifts. ADA combines the adversarial discriminative domain adaptation (ADDA) framework (Tzeng et al., 2017) with knowledge distillation.

To adapt our model to the target languages, we utilize unlabeled French, German, and English tweets from the VaccinEU dataset. Our adaptation process comprises three key steps, described as follows: (1) In Step 1, the multilingual encoder and stance classifier are fine-tuned on labeled stance data.

(2) In Step 2, the target language encoder is adapted through adversarial training and distillation. This involves training the target encoder and discriminator alternately in a two-player game. The discriminator learns to distinguish between English and non-English representations, while the target encoder learns to confound the discriminator. Knowledge distillation serves as a regularization technique, preserving information learned from the English training data.

(3) Finally, in Step 3, the adapted target encoder and stance classifier are utilized to predict stance labels for data in the target languages.

We release the code used to train and evaluate our models on GitHub (https://github.com/amcs18pd05/MTAB-cross-lingual-vaccine-stance-detection-2).

Experiments

In our experiments, we evaluate three fundamental approaches using multilingual BERT, both as standalone models and in combination with language adversarial learning. Initially, we utilize the labeled training data to train a stance classifier and a task-specific multilingual encoder. Subsequently, we adapt the encoder to the target languages French, German, and Italian by leveraging unlabeled data from the VaccinEU dataset. The Generator used for our adversarial learning experiments is the encoder for the target languages. Our primary focus in these experiments is to assess the effectiveness of our proposed MTAB that leverages adversarial learning and translation augmentation and to see its impact on the performance of zero-shot cross-lingual models.

Models

We compare a range of models that we test against our proposed MTAB model. In the absence of existing models for zero-shot cross-lingual stance detection, we use a strong, general-purpose baseline model, multilingual BERT (mBERT). In addition, we also compare with an ablated version of our MTAB model, which we refer to as MTAB-noTL. We describe next the models we compare in our experiments: mBERT—Baseline model: This model involves fine-tuning a multilingual instance of the BertForSequenceClassification model from HuggingFace on English training data. It serves as our baseline model for stance detection.

MTAB: This is our model proposed above in “MTAB: Model Architecture”. It contains the combined English vaccine stance dataset along with translations of the English training set into French, German, and Italian.

MTAB-noTL: This is an ablated version of our proposed MTAB, which does not make use of translation-augmented data and therefore could be possibly limited for the capabilities of the adversarial learning component. This model in turns tests the contribution of our proposed adversarial learning component on top of the translated data.

For each of the three models above, we test two different architectures, with and without the adversarial learning component, which we indicate in the name of the model by either including or not ‘+ Adversarial learning’ at the end of it. This leaves us with a total of six model variants, i.e., the three models listed above with and without adversarial learning component.

Model hyperparameters

We experiment with various sets of hyperparameters for our models on the validation data, and selected the ones that yielded the best results. Across all our experiments, we maintained a maximum sequence length of 128 and a batch size of 32. However, the learning rate and the number of epochs varied among the three different architectures.

In the case of MTAB, we conducted training for nine epochs using an Adam Optimizer. We experimented with learning rates of 1e−5, 2e−5, 3e−5, 4e−5, and 5e−5 and settled at 5e−5, which resulted in best scores on the validation set. This learning rate was used for the joint training of the language encoder and stance classifier. For the adversarial learning setup in MTAB, again we used an AdamW Optimizer. The learning rate was 1e−5 for both the Discriminator and the Generator, and each of the target languages went through five training epochs.

Analysis of results

Table 2 presents the results of our experiments using F1-scores. A detailed examination of the results offers various insights into the effects of model architecture on the language embeddings and subsequently the classifier performance. The combination of MTAB and adversarial language adaptation in the presence of translation augmented data emerges as the top-performing model, showcasing exceptional performance across the three target languages. This underscores the effectiveness of translation augmentation coupled with adversarial learning in achieving robust cross-lingual transfer. Interestingly, the MTAB model leveraging adversarial learning outperforms all other models consistently across the three target languages, and consequently also on average across all three languages; our full MTAB model with adversarial learning (0.520) outperforms the ablated variant with no translations by an absolute 1% (0.510) and the ablated variant with no adversarial training by 2.3% (0.497). Having looked at the overall performance scores, we next delve into different aspects of the model to better understand how this improvement happens.

Table 2 Results of experiments (F1-scores).

The entries highlighted in bold indicate the best scores for each language.

Model	VaccinEU—French	VaccinEU—German	VaccinEU—Italian	Average	
mBERT	0.50	0.50	0.46	0.487	
mBERT + AdvL	0.16	0.51	0.38	0.350	
MTAB-noTL	0.45	0.51	0.39	0.450	
MTAB-noTL + AdvL	0.51	0.55	0.47	0.510	
MTAB	0.51	0.53	0.45	0.497	
MTAB + AdvL	0.52	0.56	0.48	0.520	
Note:

AdvL, Adversarial learning.

Impact of model architecture: The decision to split the mBERT baseline model into separate models for language encoding and classifier models, as implemented in MTAB, yields notable benefits. This separation allows for higher control and flexibility over the model’s parameters. Our experiments and analysis reveal that the adversarial training in mBERT + adversarial learning, which simultaneously updates both the language encoder and classifier parameters, is very unstable and adversely impacts performance, especially for the French test set. In contrast, MTAB-noTL + adversarial learning follows a different approach—it utilizes a modified version of mBERT where the language encoding and classifier components are separated. Unlike mBERT + adversarial learning, which updates all parameters at once, MTAB-noTL + adversarial learning only updates the parameters of the target language encoder while keeping the classifier parameters fixed. This structural difference significantly reduces instability and leads to improved performance.

Effectiveness of adversarial learning: In both MTAB-noTL and MTAB, the addition of an adversarial language adaptation component contributes significantly to the F1-scores across the three target languages in the test set. As demonstrated in several other research works, this highlights the positive impact of adversarial learning on cross-lingual generalization. As stated above, however, the contribution of the adversarial learning component is notable when this is carefully integrated into the MTAB model, but fails to contribute to the baseline MBERT model.

Simple efficacy of translation augmentation: The simple and straightforward technique of augmenting training data with translations of the English tweets, as employed in MTAB, proves remarkably effective in enhancing the test scores in the absence of labelled data for the target languages. This addition surpasses the performance of MTAB-noTL both with and without an adversarial learning component.

The findings suggest that while adversarial learning enhances cross-lingual generalization, translation augmentation further amplifies this effect. This combination shows potential for handling cross-lingual tasks, especially in low-resource languages with no labeled examples.

Effect of language: A closer examination of model performance in the target languages-French, German, and Italian reveals variations in the class-wise F1-scores and shows the influence of language on stance detection. Figure 2 provides a breakdown of these scores on the test sets for the three stance labels.

Figure 2 Breakdown of the MTAB model performance showing class-wise F1-scores for the French test set.

Across all languages, models consistently perform best in the positive class. This comes from a combination of class imbalance (a higher number of positive instances in the training data) and the relative simplicity or clarity of the expression of positive sentiments across languages. In contrast, the Negative class results in the lowest F1-scores irrespective of the model or language. This is likely due to subtleties in negative expressions, such as implicit or sarcastic expressions. The performance of neutral stance varies moderately between languages, with the Italian language showing relatively strong results in this category. German stands out with generally higher F1-scores, particularly in the Positive and Neutral classes, where MTAB + AdvL achieves F1-scores near 0.80 and 0.55, respectively. French exhibits moderate performance, while Italian shows a greater balance between scores in the Neutral and Positive classes but a drop in the Negative class. However, the MTAB + AdvL delivers the highest or near-highest F1-scores across all stance labels and languages.

However, there is a considerable imbalance among the stance classes across all test languages as shown in Tables 1a and 1b. This imbalance directly affects model performance, especially on the minority classes, Neutral and Negative. Figure 2 shows the F1-scores of the models across the three stance labels: Positive, Negative and Neutral. There is a clear imbalance in the F1-scores with a difference of 10–20% between the minority classes Negative and Neutral and the majority class Positive, given the dominance of the positive class in the data. This difference is particularly more pronounced for the German language with the F1-scores for the positive label reaching as high as 80% and for minority classes around 50%. This overfitting can also be understood from the Distribution of incorrect predictions shown in Fig. 3. The figures show a consistent pattern where, in more than 50% of the incorrect predictions, the Negative and Neutral classes are incorrectly classified as Positive. To mitigate the effects of class imbalance, we explored several techniques: (1) Undersampling the majority class to balance the dataset led to reduced overall performance due to loss of informative examples. (2) Weighted loss functions slightly improved performance in the minority classes by 1%, but further tuning yielded diminishing returns. (3) Data augmentation strategies—both class-specific and uniform—were also tested, but did not yield improvements over the baseline. Although our current mitigation strategies did not yield meaningful gains, these results motivate future work in developing more sophisticated imbalance-handling techniques tailored to multilingual stance detection. These include other methods such as focal loss functions, meta-learning approaches or synthetic data preparation. We consider developing such techniques an important direction for future research.

Figure 3 Confusion matrix showing only the incorrect predictions made by the MTAB model on the French test set.

While our proposed model, MTAB, contributed with the first zero-shot cross-lingual stance detection approach and demonstrates improved performance over competitive baselines, addressing class imbalance—especially as it varies across languages—remains a critical challenge. We believe that future work should place greater emphasis on this aspect to further strengthen model robustness.

Conclusion

While prior work in cross-lingual stance detection has predominantly focused on few-shot learning scenarios, our work is, to the best of our knowledge, the first to address the zero-shot cross-lingual stance detection setting—a significantly more challenging and practically relevant scenario where no labeled training data is available in the target language. Through the introduction of the novel model called multilingual translation-augmented BERT, our research represents the first effort in zero-shot cross-lingual stance detection. Our proposed model MTAB makes the most of labeled data available from a resource-rich language, such as English, to enable the development of stance detection systems for less-resourced languages with very minimal manual effort. In doing so, we also contribute to the progression of multilingual vaccine stance detection methodologies. Through experimentation with a range of datasets labeled for stance around vaccine hesitancy and support, we have explored the possibility of transferring knowledge from a resource-rich language such as English, to other languages for which fewer and smaller labeled datasets are available, which we tested with French, German and Italian. Comparing our model with a competitive baseline model, mBERT, as well as ablated variants of our model, we demonstrate the potential of MTAB to improve cross-lingual transfer for stance detection in zero-shot settings. Further delving into the contributions of different components of the model, we also observe the importance of leveraging translation-augmented data, as well as using the adversarial learning component, for improved performance, which all in all demonstrates the effectiveness of integrating all the components of MTAB.

While our primary focus remains on leveraging deep learning approaches, we believe that future work in the area of online stance detection can benefit greatly from recent progress in the development of large language models (LLMs). We also aim to overcome the limitations of an imbalanced training dataset by exploiting the few-shot learning capabilities of LLMs enabled via prompt engineering. The development of a fine-tuned language model can also help understand the subtle nuances present in the languages used within online social media pertaining to vaccines and discussions related to public health.

Additional Information and Declarations

Competing Interests

Arkaitz Zubiaga is an Academic Editor for PeerJ.

Author Contributions

Bharathi A. conceived and designed the experiments, performed the experiments, analyzed the data, performed the computation work, prepared figures and/or tables, authored or reviewed drafts of the article, and approved the final draft.

Arkaitz Zubiaga conceived and designed the experiments, analyzed the data, authored or reviewed drafts of the article, and approved the final draft.

Data Availability

The following information was supplied regarding data availability:

The source code is available at GitHub and Zenodo:

- https://github.com/amcs18pd05/MTAB-cross-lingual-vaccine-stance-detection-2.

- A, B., & Zubiaga, A. (2025). Zero-Shot Cross-lingual Stance Detection via Adversarial Language Adaptation. Zenodo. https://doi.org/10.5281/zenodo.15547958.

The Vaccine Attitude Dataset (VADet) is available at GitHub: https://github.com/somethingx1202/VADet.

The COVID-19 vaccination tweets dataset is available at GitHub: https://github.com/a-almadan/covid-19-vaccination.

The COVID-19 Vaccine Stance dataset is available at GitHub: https://github.com/liviucotfas/covid-19-vaccination-stance-detection.

The VaccinEU (Test dataset) is available at GitHub: https://github.com/DataSciencePolimi/VaccinEU.

The model checkpoints are available at figshare: A, Bharathi; Zubiaga, Arkaitz (2025). Model checkpoints for MTAB. figshare. Online resource. https://doi.org/10.6084/m9.figshare.28941398.v1.

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
