# Peer review of "Zero-shot cross-lingual stance detection via adversarial language adaptation"

_PeerJ Computer Science, doi:10.7717/peerj-cs.2955_

## Round 0.1 · original submission · Major Revisions

Dear Authors,

Thank you for submitting your article. Based on the reviewers' comments, your article has not yet been recommended for publication in its current form. However, we encourage you to address the concerns and criticisms of the reviewers and to resubmit your article once you have updated it accordingly. Reviewer 2 has asked you to provide specific references. You are welcome to add them if you think they are usefu land relevant. However, you are under no obligation to include them, and if you do not, it will not affect my decision.

Best wishes,

Reviewer 1 ·

Basic reporting

Overall, the article makes a valuable contribution to the topics of language adaptation and zero-sale learning. With some minor editing, I think it will be publishable
1-The article is generally written in good English, but the readability would be improved if some sentences in the Introduction were more fluent.
2-The introduction sets up the relevance of the study to the existing literature well. However, the context would be stronger if more recent work on zero-shot learning and language adaptation were included.
3-The paper is structurally well organized. Figure 1 in particular clearly summarizes the model architecture. However, it would be more understandable for the reader if the terms used in tables and graphs were clearly defined.
4-The conclusion is presented in a manner consistent with the hypotheses of the study. However, further clarification could be made on the impact of the Model's increase in zero-sales performance
5- The methods are described in detail, but it would be better if the training parameters (e.g. learning rate, number of epochs) were specified. It would also be useful to provide more information about the data augmentation phase with translation
6-Detailed data analysis is a positive feature, but the criteria used for statistical significance (e.g. F1 score) could be made more descriptive

Experimental design

The article deals with issues that overlap with the areas specified in the Aims and Scope section of the journal. In particular, it makes an original contribution in the field of zero-shot language adaptation
-The research question is clearly defined and articulates the purpose of the study. The study fills an important knowledge gap in the field of language adaptation and stance detection in low-resource languages
-The paper identifies a knowledge gap in the area of zero-shot learning and language adaptation. This gap is addressed in a way that is consistent with the objectives of the study.
-The paper is of high quality, both in terms of technical details and ethical standards. Ethical use of data and model evaluation are carefully considered.
-The methods are clear and sufficiently detailed. In particular, the model architecture and data processing steps are meticulously explained, thus ensuring the reproducibility of the study.

Validity of the findings

-The paper offers an innovative perspective in the field of language adaptation and zero-shot learning. The emphasis on the replicability of the study is significant; this research can provide a solid foundation for further studies.
-The datasets used in the study are detailed and statistically reliable. The data are adequately controlled and provide a solid basis to support the research conclusions
-The results are clearly articulated to answer the initial research question. They are limited to the results obtained throughout the study, which increases the credibility of the results.

Additional comments

1- Figures and tables are very useful, but the clarity of the data could be improved if the colors used in Figure 2 were made more contrasting. In addition, in Table 3 it would be good if each column heading was clearly identified.
2-Detailing the hyperparameters in the training process of the model would contribute to easier replication of the study by other researchers.
3-The results offer an innovative perspective when compared to findings from similar studies in the literature. Especially the accuracy of the approach developed for zero-shot learning is very promising
4-If some of the concepts discussed in the introduction (e.g., adversarial language adaptation) are explained in more detail, readers unfamiliar with the topic may have a better understanding of the study.
5-The findings of this study may open the door to important applications in areas such as language adaptation and stance detection. It may be recommended to test this approach, especially for low-resource languages, in other domains

Cite this review as

Reviewer 2 ·

Basic reporting

no comment

Experimental design

no comment

Validity of the findings

no comment

Additional comments

Summary:
This paper addresses the task of zero-shot cross-lingual stance detection by proposing a Multilingual Translation-Augmented BERT (MTAB) framework. The framework employs translation augmentation to enhance the performance of cross-lingual classifiers in the absence of explicit training data in the target language. Furthermore, it integrates adversarial learning to improve the model's efficiency and performance. The authors conduct experiments on datasets in English, German, French, and Italian, demonstrating that MTAB achieves improved results compared to baseline models.

Strengths:
1. The paper is well-structured, the proposed method is straightforward, and the code is open-sourced, making it easy to follow.

2. The idea of applying adversarial domain adaptation in cross-lingual tasks is interesting, and it leads to a significant performance improvement.

Weaknesses:
1. The paper should provide a more detailed introduction to the task of zero-shot cross-lingual stance detection to help readers better understand the task and assess the paper's contribution to the field. Although the task is briefly explained in Lines 47-49 and Lines 150-151, the description is vague, which could lead to doubts about the proposed method.

2. The MTAB framework includes a Translation Augmentation process, where English training data is translated into target languages and then used as part of the training data. The motivation for this approach is confusing. In Lines 150-151, the authors state that in zero-shot cross-lingual stance detection, "there is no labeled data at all for the target language." This implies that labeled data in the target language should not appear in the training set. Additionally, I noticed that other works on cross-lingual stance detection, such as [1][2][3], either use only unlabeled target language data or only a small amount of labeled target language data.

However, Translation Augmentation introduces a large amount of labeled target language data, which clearly violates this principle. I find this approach opportunistic and even akin to cheating. If such methods are considered acceptable for zero-shot cross-lingual stance detection, I believe the task itself loses its relevance, as Translation Augmentation essentially reduces it to a "language-specific stance detection" task.

3. The experimental section requires further improvement. First, the experiments involve only a single dataset, which is insufficient. Additional datasets, such as [4], should be included to validate the robustness of the proposed method. Second, there is a lack of comparison with other baselines. I recommend adding results from similar works, such as [1][3], as strong baselines for comparison.

4. The analysis of experimental results is inadequate. In Lines 344-346, the authors mention that mBERT + AdvL training is unstable, yet the results for MTAB-noTL + AdvL in Table 2 are good. Given that MTAB's backbone is also mBERT, the results for MTAB-noTL + AdvL should align with those of mBERT + AdvL. Please clarify this discrepancy.

5. The writing quality needs improvement, and the citation format is inconsistent. For example, in Lines 118-122, the phrasing "Jamadi Khiabani and Zubiaga (2023) introduced a model that ... to improve a model for cross-target stance detection (Khiabani and Zubiaga, 2023)" is confusing and disrupts readability.

[1] Adversarial Topic-Aware Memory Network for Cross-Lingual Stance Detection (Zhang et al., IEEE ISI)\
[2] Contrastive Language Adaptation for Cross-Lingual Stance Detection (Mohtarami et al., EMNLP 2019)\
[3] Few-Shot Cross-Lingual Stance Detection with Sentiment-Based Pre-training (Hardalov et al., AAAI 2022)\
[4] X-Stance: A Multilingual Multi-Target Dataset for Stance Detection (Vamvas and Sennrich, arXiv)

Cite this review as

---

## Round 0.2 · Major Revisions

Dear Authors,

One of the previous reviewers did not respond to the invitation for reviewing the revised paper. According to the Reviewer 1, your paper still needs improvement. Please address the comments and concerns before resubmitting your paper.

Best wishes,

Reviewer 1 ·

Basic reporting

While the manuscript presents a novel architecture (MTAB) for zero-shot cross-lingual stance detection, several shortcomings limit its overall impact. Although the English is generally clear and the article follows a professional structure with relevant figures and references, there is a lack of formalism in presenting the core components. For example, key terms such as “adversarial adaptation” and “translation augmentation” are not rigorously defined, and the model components are not supported by formal proofs or theoretical justifications. Additionally, while datasets and raw data are shared, the methodology lacks detailed validation techniques—especially regarding the hyperparameter selection and generalizability of results. The absence of per-language error analysis beyond F1-scores also limits the interpretability of findings. A more thorough examination of class imbalance, domain drift, and model robustness is necessary to support the strong claims made about cross-lingual performance.

Experimental design

The manuscript aligns with the general aims of the journal and presents a relevant and timely research question in the domain of cross-lingual stance detection. However, while the problem of zero-shot stance detection is acknowledged as a gap, the justification of why this particular model (MTAB) is uniquely suited to address it could be more explicitly stated. Moreover, although the overall methodology is technically sound and based on recent deep learning practices, the experimental design lacks clarity on critical aspects such as the rationale for chosen hyperparameters, the size and composition of validation splits, and measures taken to ensure reproducibility. The methods are described at a high level but require additional implementation-level detail to allow full replication by external researchers. Providing access to configuration files, random seeds, or model checkpoints would greatly enhance transparency and replicability.

Validity of the findings

While the paper avoids overstating its impact or novelty, it would benefit from a clearer articulation of how the proposed approach meaningfully contributes to or extends prior work in cross-lingual stance detection. The availability of underlying data is commendable, and the datasets used are appropriate and statistically robust, though a deeper analysis of class imbalance effects could strengthen the statistical rigor. The conclusions are generally well aligned with the stated research objectives and are appropriately constrained to the observed results, avoiding unwarranted generalizations. However, the potential for replication would be improved with greater detail on model training procedures and evaluation protocols.

Cite this review as

---

## Round 0.3 · accepted · Accept

Dear Authors,

It is evident that the paper has undergone a substantial improvement, and the reviewer has accepted it.

Best wishes,

Reviewer 1 ·

Basic reporting

Thank you for sharing the manuscript titled “Zero-shot Cross-lingual Stance Detection via Adversarial Language Adaptation” (Revision #2). Based on the PeerJ criteria, here is a structured and professional review that reflects the improvements made during revision and assesses the current state of the manuscript.

Experimental design

The manuscript presents original primary research that fits well within the aims and scope of the journal, addressing a relevant and timely challenge in multilingual natural language processing. The research question is clearly defined and focuses on the problem of zero-shot cross-lingual stance detection—a scenario where no labeled data is available for the target language. This is a meaningful and practical problem, especially in low-resource settings, and the authors convincingly state how their proposed approach, MTAB (Multilingual Translation-Augmented BERT), fills this gap. The investigation is conducted to a high technical and ethical standard, with the use of publicly available datasets, compliance with Twitter's data-sharing policies, and a transparent experimental pipeline. The methodology is described in sufficient detail to allow replication, including clear documentation of preprocessing steps, model components, training procedures, and hyperparameter choices. Furthermore, the availability of the source code enhances the reproducibility and integrity of the study. Overall, the study demonstrates methodological rigor, originality, and relevance to the field.

Validity of the findings

The manuscript meets the validity criteria expected for publication. While the impact and novelty of the work are not formally assessed, the study encourages meaningful replication by clearly outlining the rationale and situating the research within a well-defined gap in the literature—namely, the lack of zero-shot cross-lingual stance detection models. The authors provide all underlying data through processed tweet IDs and open-source code, ensuring transparency and reproducibility. The datasets used are robust and appropriately controlled, and the experimental results are statistically sound, supported by class-wise F1-scores, confusion matrices, and ablation studies. The conclusions are clearly articulated, logically derived from the findings, and directly tied to the research objectives. They are appropriately limited to what the results support, avoiding overgeneralization. Overall, the findings are valid and reliably presented.

Additional comments

none

Cite this review as